# Risk Factors for Cervical Lymph Node Metastasis in Middle Eastern Papillary Thyroid Microcarcinoma

**DOI:** 10.3390/jcm11154613

**Published:** 2022-08-08

**Authors:** Sandeep Kumar Parvathareddy, Abdul K. Siraj, Padmanaban Annaiyappanaidu, Nabil Siraj, Saif S. Al-Sobhi, Fouad Al-Dayel, Khawla S. Al-Kuraya

**Affiliations:** 1Human Cancer Genomic Research, King Faisal Specialist Hospital and Research Centre, P.O. Box 3354, Riyadh 11211, Saudi Arabia; 2Department of Surgery, King Faisal Specialist Hospital and Research Centre, P.O. Box 3354, Riyadh 11211, Saudi Arabia; 3Department of Pathology, King Faisal Specialist Hospital and Research Centre, P.O. Box 3354, Riyadh 11211, Saudi Arabia

**Keywords:** papillary thyroid microcarcinoma, lymph node metastasis, recurrence-free survival, risk factors

## Abstract

Papillary thyroid microcarcinoma (PTMC) typically has an indolent course and excellent prognosis. Nonetheless, a subset of PTMC carries a risk of lymph node metastasis (LNM) and local recurrence. PTC from the Middle Eastern population is unique with respect to demographic and clinico-pathological characteristics as compared to other ethnicities of the world. The risk factors of LNM in PTMC patients of Middle Eastern ethnicity have not been fully explored. The present study aims to investigate the influencing factors of LNM in Middle Eastern PTMC patients and its predictive impact on patient’s outcome. A total of 226 confirmed PTMC cases were selected in this retrospective study. The correlation between clinico-pathological, as well as molecular, characteristics and LNM was evaluated. Multivariate analysis was performed by logistic regression and Cox proportional hazards models. Among the 226 patients, the rate of LNM was 43.8% (99/226). Bilaterality, multifocality, gross extrathyroidal extension (ETE), and intermediate-to-high American Thyroid Association (ATA) risk tumors were significantly associated with LNM in PTMC. Multivariate logistic regression analysis showed that bilaterality and gross ETE were independent predictive factors for LNM in PTMC. The recurrence-free survival (RFS) was shorter in PTMC with LNM compared to those without LNM (*p* = 0.0051) and was significant on multivariate analysis. In conclusion, our study showed that bilaterality and gross ETE were independent influencing factors of LNM in Saudi patients with PTMC. LNM was also associated with shorter RFS. The identification of risk factors for LNM in patients of Middle Eastern ethnicity could help the individualization of clinical management for PTMC patients.

## 1. Introduction

The incidence of papillary thyroid carcinoma (PTC) has increased in the past decades [1,2,3]. In Saudi Arabia, thyroid cancer is the second most common cancer in women [4]. Papillary thyroid microcarcinoma (PTMC) is defined by the World Health Organization (WHO) as a tumor measuring up to 10 mm in its largest dimension [5]. The incidence of PTMC has also increased, primarily due to improved detection methods such as high-resolution sensitive cervical ultrasound and subsequent use of fine needle aspiration (FNA) biopsy [6,7,8,9]. Despite the increasing incidence of PTMC, it is indolent in nature, with patients usually having favorable prognosis [10]. However, a considerable subset of patients with PTMC have a tendency to develop aggressive disease and lymph node metastasis (LNM) [11,12].

According to several reports, LNM is considered one of the risk factors for aggressive PTMC [10,13,14,15,16]. Several clinico-pathological and molecular characteristics have been shown to increase the risk for LNM such as older age, gender, multifocality, bilaterality, extra-thyroidal extension, and the presence of Hashimoto’s thyroiditis [15,17,18]. In addition, the presence of *BRAF* mutations has been reported to be a significant risk factor for LNM [19,20].

Furthermore, the rate of LNM varies in frequency from one study to another, ranging between 14% to 40% [21,22,23]. Chow et al. [24] and others [25,26] have also documented the negative impact of LNM on recurrence of PTMC patients. However, at present, there are limited studies on the risk factors for LNM in PTMC in Middle Eastern populations. PTC in those of Middle Eastern ethnicity differs from other populations of the world with respect to several demographic and clinico-pathological features. The incidence of PTC is higher in Saudi Arabia compared to Western populations and it ranks as the second most common malignancy among females in this ethnicity, accounting for 14.3% of all cancers [4]. Further, as reported by this study and in other studies on Middle Eastern populations, the incidence of recurrence (18.4%) [27], lymph node metastasis (53.0%) [28], and distant metastasis (7.5%) [29] is higher compared to other ethnicities. Another important difference is the earlier age of onset for PTC in Middle Eastern populations. The median age of our study population is 40.4 years, which is similar to studies of other Middle Eastern ethnicities [30,31,32,33], but lower than that seen in studies of Western populations [34]. These findings likely reflect the inherent aggressive nature of PTC in this ethnicity.

Therefore, there is a compelling need to explore LNM in PTMC in this ethnicity to improve patient management. This study aims to explore the rate and clinico-pathological and molecular risk factors, as well as the impact of LNM, on patient outcomes in Middle Eastern PTMC.

## 2. Materials and Methods

### 2.1. Patient Selection

A total of 226 PTMC patients diagnosed between 1988 and 2018 at King Faisal Specialist Hospital and Research Centre (Riyadh, Saudi Arabia) were included in the study. Cases were identified based on clinical history, followed by FNA biopsy for confirmation. The Institutional Review Board of the hospital approved this study, and since only retrospective patient data were used, the Research Advisory Council (RAC) provided a waiver of consent under project RAC # 221 1168 and # 2110 031. The study was conducted in accordance with the Declaration of Helsinki.

### 2.2. Clinico-Pathological and Follow-Up Data

Baseline clinico-pathological data were collected from case records and have been summarized in Table 1. A total thyroidectomy was performed in 81.4% (184/226) of PTMC cases in our cohort, whereas 18.6% (42/226) underwent lobectomy/hemithyroidectomy. Prophylactic central lymph node dissection was only performed in patients with clinically uninvolved central neck lymph nodes (cN0) who had advanced primary tumors (T3 or T4) or clinically involved lateral neck nodes (cN1b), or, if the information could be used to plan further steps in therapy, in accordance with the 2015 American Thyroid Association (ATA) guidelines [35]. Prophylactic lateral lymph node dissection was performed only if there was intra-operative evidence of lateral lymph node involvement. Extra-thyroidal extension (ETE) was further classified as follows: microscopic ETE was defined as a tumor extending beyond the thyroid capsule into the surrounding peri-thyroidal soft tissues of fat and/or skeletal muscle, without visual evidence of this invasion, and macroscopic ETE was defined as visual evidence of tumor invasion into strap muscles, subcutaneous soft tissue, larynx, trachea, esophagus, recurrent laryngeal nerve, or prevertebral fascia. Based on the 2015 ATA guidelines, tall cell, hobnail, columnar cell, diffuse sclerosing, and insular variants were classified as aggressive variants, whereas classical and follicular variants were classified as non-aggressive variants [35]. The staging of PTMC was performed using the eighth edition of American Joint Committee on Cancer (AJCC) staging system. Patients were stratified into low, intermediate, and high risk based on the 2015 ATA guidelines [35]. Following initial surgery, low-risk PTC patients were followed up annually, intermediate-risk patients were followed up at 6 months intervals, and high-risk patients were followed up at 3 months intervals. At each follow-up, neck ultrasound, thyroid function tests, thyroglobulin levels, and thyroglobulin antibodies were performed. In addition, for high-risk patients, radioiodine scans and PET CT scans were performed to identify tumor recurrence. Only structural recurrence (local, regional, or distant) was considered for analysis. Recurrence was defined as any newly detected tumor (local or distant) or metastatic regional lymph node based on ultrasound and/or imaging studies in patients who had been previously free of disease following initial treatment. Recurrence-free survival (RFS) was defined as the time (in months) from the date of initial surgery to the occurrence of any tumor recurrence (local, regional, or distant). In the case of no recurrence, the date of the last follow-up was the study endpoint for RFS.

### 2.3. BRAF and TERT Mutation Analysis

*BRAF* and *TERT* mutation data for the entire PTMC cohort were available from our previous studies [36,37].

### 2.4. Statistical Analsysis

The associations between clinico-pathological variables and tumor recurrence were determined using contingency table analysis and Chi-square tests. Logistic regression analysis was used for multivariate analysis. The Mantel–Cox log rank test was used to evaluate RFS. Survival curves were generated using the Kaplan–Meier method. Two-sided tests were used for statistical analyses with a limit of significance defined as a *p* value of <0.05. Data analyses were performed using the JMP14.0 (SAS Institute, Inc., Cary, NC, USA) software package.

## 3. Results

### 3.1. Patient Characteristics

The median age of the study cohort was 40.4 years (range: 11.5–84 years), with a male to female ratio of 1:4. A majority of the tumors were non-aggressive variants (classical variant–71.2%; 161/226, follicular variant–16.0%; 36/226). Tumors were bilateral in 31.9% (72/226) of cases, whereas multifocality was noted in 40.3% (91/226). Extrathyroidal extension was seen in 27.9% (63/226) of PTMCs. The incidence of *BRAF* mutation in PTMC was 46.8% (103/220) and that of *TERT* mutation was 8.0% (16/200). On ATA risk stratification, 29.6% (67/226) were found to be low risk, 40.8% (92/226) intermediate risk, and 29.6% (67/226) high risk. Of the patients, 72.6% (164/226) received radioactive iodine therapy (Table 1).

### 3.2. Incidence of Lymph Node Metastasis (LNM) in PTMC and Its Clinico-Pathological Associations

LNM was noted in 43.8% (99/226) of PTMCs in our cohort, of which central LNM (CLNM) was 17.7%, and lateral LNM (LLNM) was 26.1%. The rate of occult CLNM (clinical N0 with pathologic N1a) was 5.3% and the rate of occult LLNM (clinical N0 with pathologic N1b) was 7.5%. A total of 24 (10.6%) patients had Nx status and, hence, were not included for further analysis. LNM was significantly associated with bilateral tumors (*p* < 0.0001), multifocality (*p* = 0.0019), gross extrathyroidal extension (*p* = 0.0034), tumor recurrence (*p* = 0.0025), and ATA intermediate/high risk tumors (*p* < 0.0001). However, no association with molecular markers (*RAF/RAS* genes and *TERT*) was noted (Table 2).

### 3.3. Risk Factors Predicting LNM in PTMC

Since we found LNM to be associated with adverse clinico-pathological factors in PTMC, we sought to further assess the risk factors that could independently predict LNM. On multivariate logistic regression analysis, we found gross extrathyroidal extension (HR = 3.20; 95% CI = 1.36–7.53; *p* = 0.0077) and tumor bilaterality (HR = 3.31; 95% CI = 1.43–7.62; *p* = 0.0050) to be independent predictors for LNM (Table 3).

### 3.4. LNM and Clinical Outcome in PTMC

As previously stated, LNM was significantly associated with tumor recurrence. Using the Mantel–Cox log rank test, we found that PTMC patients with LNM had a significantly shorter RFS compared to patients without LNM (*p* = 0.0051; Figure 1). On multivariate analysis using Cox proportional hazards model, LNM was found to be an independent predictor of shorter RFS (HR = 2.97; 95% CI = 1.23–7.99; *p* = 0.0149) (Table 4).

## 4. Discussion

PTMC has become a health concern due to its increase in incidence over recent decades [19,20]. Although cases of PTMC usually have favorable outcomes, there is a subset of PTMC patients that present with aggressive disease and a high risk of recurrence. Identifying risk factors of LNM and recurrence is very important to decide the best therapeutic management for these patients.

In the present study, we analyzed the retrospective data of 226 patients with PTMC. The results revealed that the rate of LNM was 43.8%, which is within the range previously reported [15,38]. The rate of CLNM was 17.7% and LLNM was 26%. Interestingly, the rate of occult CLNM was 5.3%, whereas it was 7.5% for occult LLNM. This suggests that lateral LNs are more commonly involved, especially for occult metastasis. This is different from previous studies, where central LNs were the most common area for LNM [15,21,39]. This result of a high rate of LLNM and occult LLNM could help surgeons in the decision of performing prophylactic lateral lymph node dissections to reduce recurrence. However, a larger cohort from this ethnic group is required to validate our findings.

This study shows the presence of bilateral foci and gross extra-thyroid extension as the only independent risk factors for LNM in Middle Eastern patients with PTMC. This is similar to other studies which also found extrathyroidal extension and bilaterality as predictors of lymph node metastasis [14,17,21,40]. However, our study differs from larger studies of other Western and far Eastern ethnicities in that we did not find any other clinico-pathological parameters as potential predictors of lymph node metastasis, whereas male gender, age, tumor size, multifocality, lymphovascular invasion, and presence of Hashimoto’s thyroiditis were reported in these ethnicities as predictors of lymph node metastasis in PTMC [14,15,17,21,40,41,42,43,44,45,46,47]. The cohort size, ethnicity, and study design (including lateral LNM or central LNM) might contribute to these differences.

The presence of bilateral foci has been reported in the literature as an independent risk factor for LNM [23,48]. Although gross ETE has not been reported as an independent risk factor of LNM in patients with PTMC, its strong association with aggressive PTMC and its impact on patient survival have been well documented [10,49,50].

Furthermore, we found that the proportion of intermediate/high ATA risk of recurrence stratification is significantly higher in LNM PTMC patients, which suggests a higher risk of recurrence. Therefore, we have investigated the impact of LNM on PTMC patient outcome. The result revealed a 2.8-fold higher recurrence rate in PTMC patients with LNM than in PTMC patients without LNM. In addition, the recurrence-free survival was significantly shorter (*p* = 0.0051) in LNM PTMC patients. This is in agreement with other previous studies, where LNM was found to be a predictor of patient outcome [51,52]. While some studies have shown that LNM can affect a patient’s recurrence-free survival only in combination with other risk factors, such as number and diameter of LNM [53,54], this study shows LNM alone to be an independent risk factor for recurrence-free survival, irrespective of the number and diameter of LNM in PTMC.

This study has some limitations. It is a retrospective study from a single tertiary care center, and so selection bias cannot be ignored. The study has a limited number of PTMC patients of Middle Eastern ethnicity, and so the conclusions need to be read with caution.

## 5. Conclusions

Bilateral tumors and gross ETE are independent risk factors for LNM in Middle Eastern PTMC patients. The presence of lateral LNM is apparently higher than central LNM in this study cohort, which should be considered carefully by surgeons for therapeutic and preventive plans for this PTMC patient group. Finally, we identified that LNM in PTMC increases the risk of recurrence and affects the prognosis by being an independent risk factor for recurrence-free survival. These data could help to extend our understanding about the risk of LNM and recurrence in PTMC in patients of Middle Eastern ethnicity and support the importance of individualized management of PTMC patients.

## Figures and Tables

**Figure 1 jcm-11-04613-f001:**
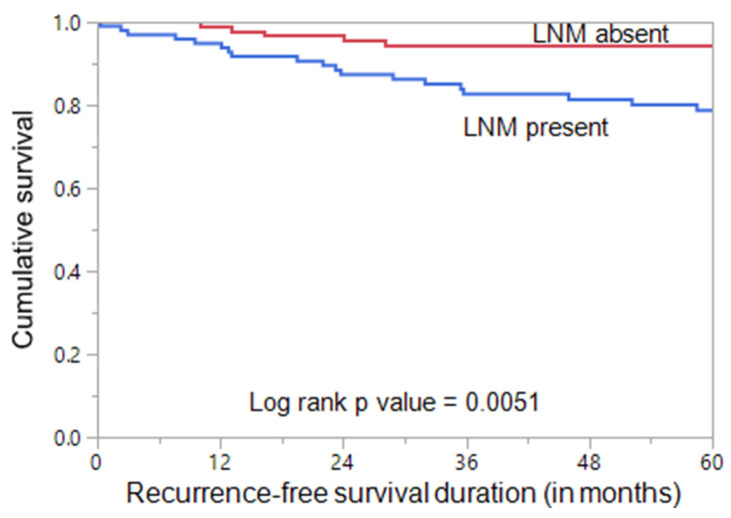
**Recurrence-free survival (RFS) in papillary thyroid microcarcinoma (PTMC).** Kaplan–Meier survival curve showing a shorter recurrence-free survival in PTMC patients with lymph node metastasis (LNM) than in those without LNM (*p* = 0.0051).

**Table 1 jcm-11-04613-t001:** Clinico-pathological and molecular characteristics of papillary thyroid microcarcinoma.

	No.	%
Total	226	
Age, median (range)	40.4 (11.5–84.0)	
<55	191	84.5
≥55	35	15.5
Gender		
Female	181	80.1
Male	45	19.9
Histologic subtype		
Aggressive variants	29	12.8
Non-aggressive variants	197	87.2
Tumor laterality		
Unilateral	154	68.1
Bilateral	72	31.9
Multifocality		
Yes	91	40.3
No	135	59.7
Extrathyroidal extension		
Gross	15	6.7
Microscopic	48	21.2
Absent	163	72.1
Lymphovascular invasion		
Present	42	18.6
Absent	184	81.4
Tumor size		
<0.5 cm	54	23.9
0.5–1.0 cm	172	76.1
pN		
N0	103	45.6
N1	99	43.8
Central only	40	17.7
Central + lateral	8	3.5
Lateral only	51	22.6
Nx	24	10.6
pM		
M0	220	97.3
M1	6	2.7
TNM Stage		
I	206	91.2
II	11	4.8
III	3	1.3
IV	4	1.8
Unknown	2	0.9
Hashimoto’s thyroiditis		
Present	46	20.4
Absent	180	79.6
*BRAF* mutation		
Present	103	45.6
Absent	117	51.8
Unknown	6	2.6
*NRAS* mutation		
Present	16	7.1
Absent	203	89.8
Unknown	7	3.1
*HRAS* mutation		
Present	3	1.3
Absent	217	96.0
Unknown	6	2.7
*KRAS* mutation		
Present	2	0.9
Absent	218	96.4
Unknown	6	2.7
*TERT* mutation		
Present	16	7.1
Absent	184	81.4
Unknown	26	11.5
Recurrence		
Yes	33	14.6
No	193	85.4
ATA risk category		
Low	67	29.6
Intermediate	92	40.8
High	67	29.6
Total follow-up duration (mean ± S.D.) (in years)	9.8 ± 6.9	

TNM—Tumor-Node-Metastasis; *BRAF*—B-Raf Proto-Oncogene; *NRAS*—Neuroblastoma RAS Viral Oncogene Homolog; *HRAS*—Harvey Rat Sarcoma Viral Oncogene Homolog; *KRAS*—Kirsten rat sarcoma virus; *TERT*—Telomerase Reverse Transcriptase.

**Table 2 jcm-11-04613-t002:** Clinico-pathological associations of lymph node metastasis in papillary thyroid microcarcinoma.

	Total		LNM Present		LNM Absent		*p* Value
	No.	%	No.	%	No.	%	
Total	202		99	49.0	103	51.0	
Age (years)							
<55	171	84.6	88	88.9	83	80.6	0.0992
≥55	31	15.4	11	11.1	20	19.4	
Gender							
Female	165	81.7	77	77.8	88	85.4	0.1587
Male	37	18.3	22	22.2	15	14.6	
Histologic subtype							
Aggressive variants	26	12.9	12	12.1	14	13.6	0.7548
Non-aggressive variants	176	87.1	87	87.9	89	86.4	
Tumor laterality							
Unilateral	132	65.3	51	51.5	81	78.6	<0.0001
Bilateral	70	34.7	48	48.5	22	21.4	
Multifocality							
Yes	84	41.6	52	52.5	32	31.1	0.0019
No	118	58.4	47	47.5	71	68.9	
Extrathyroidal extension							
None	145	71.8	61	61.6	84	81.5	0.0034
Microscopic	42	20.8	26	26.2	16	15.6	
Gross	15	7.4	12	12.2	3	2.9	
Lymphovascular invasion							
Present	35	17.3	17	17.2	18	17.5	0.9545
Absent	167	82.7	82	82.8	85	82.5	
Tumor size							
≤0.5 cm	47	23.3	24	24.2	23	22.3	0.7478
0.6–1.0	155	76.7	75	75.8	80	77.7	
Distant metastasis							
Yes	12	5.9	9	9.1	3	2.9	0.0582
No	190	94.1	90	90.9	100	97.1	
TNM stage							
I–II	195	96.5	94	94.9	101	98.1	0.2155
III–IV	7	3.5	5	5.1	2	1.9	
Hashimoto’s thyroiditis							
Yes	40	19.8	19	19.2	21	20.4	0.8310
No	162	80.2	80	80.8	82	79.6	
*BRAF* mutation							
Present	95	47.0	53	53.5	42	40.8	0.0693
Absent	107	53.0	46	46.5	61	59.2	
*NRAS* mutation							
Present	15	7.7	4	4.2	11	11.0	0.1061
Absent	180	92.3	91	95.8	89	89.0	
*HRAS* mutation							
Present	3	1.5	1	1.0	2	2.0	0.5809
Absent	193	98.5	95	99.0	98	98.0	
*KRAS* mutation							
Present	1	0.5	0	0.0	1	1.0	0.2451
Absent	195	99.5	96	100.0	99	99.0	
*TERT* mutation							
Present	15	8.5	8	9.0	7	8.1	0.8227
Absent	161	91.5	81	91.0	80	91.9	
Tumor recurrence							
Yes	33	16.3	24	24.2	9	8.7	0.0025
No	169	83.7	75	75.8	94	91.3	
ATA risk category							
Low	63	31.2	2	2.0	61	59.2	<0.0001
Intermediate/high	139	68.8	97	98.0	42	40.8	

TNM—Tumor-Node-Metastasis; *BRAF*—B-Raf Proto-Oncogene; *NRAS*—Neuroblastoma RAS Viral Oncogene Homolog; *HRAS*—Harvey Rat Sarcoma Viral Oncogene Homolog; *KRAS*—Kirsten rat sarcoma virus; *TERT*—Telomerase Reverse Transcriptase.

**Table 3 jcm-11-04613-t003:** Multivariate analysis using a logistic regression model for predictors of lymph node metastasis (LNM) in PTMC.

	Multivariate (LNM)	
Clinico-Pathological Variables	HR (95% CI)	*p* Value
Age		
≥55 years (vs. <55years)	0.44 (0.18–1.09)	0.0747
Gender		
Male (vs. female)	1.45 (0.63–3.31)	0.3794
Histology		
Aggressive variants (vs. non-aggressive variants)	0.96 (0.38–2.45)	0.9322
Tumor laterality		
Bilateral (vs. unilateral)	3.31 (1.43–7.62)	0.0050
Tumor focality		
Multifocal (vs. unifocal)	0.91 (0.40–2.06)	0.8261
Extrathyroidal extension		
Absent	Reference	
Microscopic	1.25 (0.36–4.31)	0.7240
Gross	3.20 (1.36–7.53)	0.0077
Lymphovascular invasion		
Present (vs. absent)	0.73 (0.30–1.76)	0.4804
Tumor size		
0.5–1.0 cm (vs. <0.5 cm)	0.95 (0.46–1.95)	0.8848
Distant metastasis		
Present (vs. absent)	2.44 (0.50–11.99)	0.2728

HR—hazard ratio; CI—confidence interval.

**Table 4 jcm-11-04613-t004:** Multivariate analysis using a Cox proportional hazards model for predictors of recurrence-free survival (RFS) in PTMC.

	Multivariate (RFS)	
Clinico-Pathological Variables	HR (95% CI)	*p* Value
Age		
≥55 years (vs. <55 years)	1.89 (0.57–5.19)	0.2760
Gender		
Male (vs. female)	2.92 (1.28–6.48)	0.0116
Histology		
Aggressive variants (vs. non-aggressive variants)	0.52 (0.07–2.27)	0.4238
Tumor laterality		
Bilateral (vs. unilateral)	1.21 (0.35–5.63)	0.7766
Tumor focality		
Multifocal (vs. unifocal)	0.52 (0.11–1.78)	0.3139
Extrathyroidal extension		
Absent	Reference	
Microscopic	2.40 (0.86–6.47)	0.0937
Gross	3.29 (0.70–16.72)	0.1325
Lymphovascular invasion		
Present (vs. absent)	0.58 (0.09–2.10)	0.4436
Tumor size		
0.5–1.0 cm (vs. <0.5 cm)	0.81 (0.17–3.02)	0.7658
Distant metastasis		
Present (vs. absent)	4.37 (1.08–18.16)	0.0389
Lymph node metastasis		
Present (vs. absent)	2.97 (1.23–7.99)	0.0149

HR—hazard ratio; CI—confidence interval.

## Data Availability

The data presented in this study are available in the article.

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
