# Peer review of "Risk Factors for Cervical Lymph Node Metastasis in Middle Eastern Papillary Thyroid Microcarcinoma"

_jcm, 2022, doi:10.3390/jcm11154613_

Round 1

Reviewer 1 Report

I have read with great interest the manuscript entitled “Risk factors for cervical lymph node metastasis in Middle Eastern papillary thyroid microcarcinoma”. In this study the authors investigated a cohort of 226 with PTMC and identified LNM in 44%. On multivariate analysis, risk factors for LNM were bilateral disease as well as ETE. As expected, LNM was associated with shorter RFS.

Comments –

1.     Why is this study unique to the middle east? Is your population any different than the rest of the world? In my opinion the paper would read just fine without any mention of the middle east but rather a single tertiary center experience.

2.     As per above, your data is not novel and much larger series evaluated risk factors for LNM in PTMCs (with similar results regarding ETE).

3.     Methods- what was your surgical strategy? All of your patients underwent total thyroidectomy for PTMC? Did you routinely perform prophylactic CLND on all patients? How can you have patients with occult lateral LNM? Did you perform prophylactic lateral LNM for patients with PTMC?? This is certainly not the standard of care. Your institutional surgical policy should be clearly stated.

4.     Methods – the rate of RAI belongs to the results.

5.     LNM – data gives the wrong impression that 26.1% had lateral LNM only whereas I think that most of these patients had lateral and central LNM. Did you actually have any patients with lateral LNM only? In any case, this should be clarified.

Author Response

Response to Reviewer 1 comments

I have read with great interest the manuscript entitled “Risk factors for cervical lymph node metastasis in Middle Eastern papillary thyroid microcarcinoma”. In this study the authors investigated a cohort of 226 with PTMC and identified LNM in 44%. On multivariate analysis, risk factors for LNM were bilateral disease as well as ETE. As expected, LNM was associated with shorter RFS.

Response: We thank the reviewer for taking the time to review our manuscript and for providing valuable suggestions to further improve the study. We have addressed the reviewer’s concerns point-by-point below. The reviewer’s comments are in black and our replies are colored in red. We hope the reviewer finds our replies and revisions to be satisfactory.

Comments

Point 1: Why is this study unique to the middle east? Is your population any different than the rest of the world? In my opinion the paper would read just fine without any mention of the middle east but rather a single tertiary center experience.

Response 1: We appreciate the reviewer’s concern regarding the uniqueness of Middle Eastern population. Indeed, papillary thyroid carcinoma from the Middle Eastern population differs from other ethnicities. The incidence of PTC is higher in Saudi Arabia compared to Western population and it ranks as the second most common malignancy among females in this ethnicity, accounting for 14.3% of all cancers. Also, as reported by us and other studies from Middle East, the incidence of recurrence (18.4%) (1), lymph node metastasis (53.0%) (2) and distant metastasis (7.5%) (3) is higher compared to other ethnicities. Another important difference is the earlier age of onset for PTC in the Middle Eastern population. The median age of our study population is 40.4 years, which is similar to studies from other Middle Eastern ethnicities (4-7), but lower than that seen in Western population (8). These findings probably reflect the inherent aggressive nature of PTC in this ethnicity. We wanted to emphasize that PTC from Middle Eastern ethnicity differs from other ethnicities and hence included it in the title. However, we do agree that this is a retrospective single tertiary care center study, which could lead to selection bias. We have mentioned this as a limitation of our study in the revised manuscript (Page 10, Line 258).

Point 2: As per above, your data is not novel and much larger series evaluated risk factors for LNM in PTMCs (with similar results regarding ETE).

Response 2: We fully agree with the reviewer that there have been larger studies evaluating risk factors for LNM in PTMCs. However, at present, there are limited studies on the risk factors for LNM in PTMC from Middle Eastern ethnicity. Therefore, we believe there is a compelling need to explore LNM in PTMC to improve patients’ management in this ethnicity. Our study findings add to the existing literature and could pave the way for larger studies from this ethnicity to better characterize PTMC in the Middle Eastern population.

Point 3: Methods- what was your surgical strategy? All of your patients underwent total thyroidectomy for PTMC? Did you routinely perform prophylactic CLND on all patients? How can you have patients with occult lateral LNM? Did you perform prophylactic lateral LNM for patients with PTMC?? This is certainly not the standard of care. Your institutional surgical policy should be clearly stated.

Response 3: We appreciate the reviewer’s concern regarding surgical strategy in our institute. Total thyroidectomy was performed in 81.4% (184/226) of PTMC cases in our cohort, whereas 18.6% (42/226) underwent lobectomy/hemithyroidectomy. Prophylactic CLND was only performed in patients with clinically uninvolved central neck lymph nodes (cN0) who had advanced primary tumors (T3 or T4) or clinically involved lateral neck nodes (cN1b), or if the information could be used to plan further steps in therapy, in accordance with the 2015 ATA guidelines (9).

With regards to occult lateral LNM, prophylactic lateral lymph node dissection was performed only if there was intra-operative evidence of lateral lymph node involvement. Indeed, previous studies have also evaluated the utility of prophylactic lateral lymph node dissection in PTC (10, 11).  We have now clearly stated our institutional surgical policy in the Methods section (Page 2, Line 73 – 80).

Point 4: Methods – the rate of RAI belongs to the results.

Response 4: We thank the reviewer for their suggestion. We have now incorporated the rate of RAI in the Results section of the revised manuscript section (Page 3, Line 126).

Point 5: LNM – data gives the wrong impression that 26.1% had lateral LNM only whereas I think that most of these patients had lateral and central LNM. Did you actually have any patients with lateral LNM only? In any case, this should be clarified.

Response 5: We appreciate the reviewer’s concern regarding rate of lateral LNM in our cohort. Of the 43.8% (99/226) cases showing lymph node metastasis, central LNM alone was noted in 17.7% (40/226), lateral LNM alone in 22.6% (51/226) and both central and lateral LNM in 3.5% (8/226) of cases. We have now clarified this in Table 1 of the revised manuscript.

References

  1. Siraj AK, Parvathareddy SK, Qadri Z, Siddiqui K, Al-Sobhi SS, Al-Dayel F, Al-Kuraya KS. Annual hazard rate of recurrence in Middle Eastern papillary thyroid cancer over a long-term follow-up. Cancers. 2020 Dec 3;12(12):3624.
  2. Parvathareddy SK, Siraj AK, Qadri Z, Ahmed SO, DeVera F, Al-Sobhi S, Al-Dayel F, Al-Kuraya KS. Lymph node ratio is superior to AJCC N stage for predicting recurrence in papillary thyroid carcinoma. Endocrine Connections. 2022 Feb 1;11(2).
  3. Ahmed SO, Siraj AK, Parvathareddy SK, Iqbal K, Qadri Z, Al-Rasheed M, Siraj S, Thangavel S, Diaz R, Benito A, Victoria IF. TERT promoter mutations are an independent predictor of distant metastasis in Middle Eastern papillary thyroid microcarcinoma. Cancer Research. 2022 Jun 15;82(12_Supplement):5280-.
  4. Doubi A, Al-Qannass A, Al-Angari SS, Al-Qahtani KH, Alessa M, Al-Dhahri S. Trends in Thyroid Carcinoma Among Thyroidectomy Patients: A 12-Year Multicenter Study. Ann Saudi Med (2019) 39(5):345–9.
  5. Samargandy S, Qari R, Aljadani A, Assaqaf D, Etaiwi A, Alghamdi D, et al.. Clinicopathological Characteristics of Thyroid Cancer in a Saudi Academic Hospital. Cureus (2020) 12(5):e8044.
  6. Al-Zaher N, Al-Salam S, El Teraifi H. Thyroid Carcinoma in the United Arab Emirates: Perspectives and Experience of a Tertiary Care Hospital. Hematol/Oncol Stem Cell Ther (2008) 1(1):14–21.
  7. Keinan-Boker L, Silverman BG. Trends of Thyroid Cancer in Israel: 1980–2012. Rambam Maimonides Med J (2016) 7(1):e0001. doi: 10.5041/RMMJ.10228.
  8. Lim H, Devesa SS, Sosa JA, Check D, Kitahara CM. Trends in Thyroid Cancer Incidence and Mortality in the United States, 1974-2013. Jama (2017) 317(13):1338–48.
  9. Haugen BR, Alexander EK, Bible KC, Doherty GM, Mandel SJ, Nikiforov YE, et al. 2015 American Thyroid Association Management Guidelines for Adult Patients with Thyroid Nodules and Differentiated Thyroid Cancer: The American Thyroid Association Guidelines Task Force on Thyroid Nodules and Differentiated Thyroid Cancer. Thyroid (2016) 26(1):1–133.
  10. Lim, Y. , Lee, J. , Lee, Y. S. , Lee, B. , Wang, S. , Son, S. & Kim, I. Lateral cervical lymph node metastases from papillary thyroid carcinoma: Predictive factors of nodal metastasis. Surgery (2011) 150 (1), 116-121.
  11. Zhan S, Luo D, Ge W, Zhang B, Wang T. Clinicopathological predictors of occult lateral neck lymph node metastasis in papillary thyroid cancer: A meta-analysis. Head & Neck. 2019; 1–9.

Reviewer 2 Report

I thank the editor and the Journal for giving me the opportunity to review this manuscript. I've read this paper with great interest.
This is a retrospective study that aim to evaluate the risk factors and the impact on patients’ outcome of lymph node metastasis in papillary thyroid microcarcinoma (PTMC). The Authors included 202 patients and found that bilateral tumours and gross extrathyroidal extension are independent risk factors for Lymph node metastasis (LNM) in PTMC in Middle Eastern population; they even identified that LNM in PTMC increases the risk of recurrence and affect the prognosis.
In my opinion, the methodology of the study is clear. The statistical analysis are accurate and exhaustive.
Discussion and conclusions of the study are presented clearly and comprehensively and are consistent with the reported results.

Herein I report only minor concerns:

- There are some typos and an English revision is required; please check the text again;

- In line 23, the acronym ATA is not made explicit previously in the text;

-In line 70, the acronym ETE is not made explicit previously in the text;

-To give uniformity to the text and make it easier to read, the Authors must report all the data in Arabic numbers, instead of spelling them.

- In my opinion, the manuscript could be enriched by these articles:

- DOI: 10.3390/jcm10184076, to be inserted in the introduction, where the references 1 and 2 are present

- DOI: 10.3390/cancers13215567, to be inserted in the introduction, where the references 17 is present

- DOI: 10.1371/journal.pone.0121514, to be inserted in the introduction, where the references 10 is present.

Author Response

Response to Reviewer 2 comments

I thank the editor and the Journal for giving me the opportunity to review this manuscript. I've read this paper with great interest.

This is a retrospective study that aim to evaluate the risk factors and the impact on patients’ outcome of lymph node metastasis in papillary thyroid microcarcinoma (PTMC). The Authors included 202 patients and found that bilateral tumours and gross extrathyroidal extension are independent risk factors for Lymph node metastasis (LNM) in PTMC in Middle Eastern population; they even identified that LNM in PTMC increases the risk of recurrence and affect the prognosis.

In my opinion, the methodology of the study is clear. The statistical analysis are accurate and exhaustive. Discussion and conclusions of the study are presented clearly and comprehensively and are consistent with the reported results.

Response: We thank the reviewer for their time and effort in reviewing our manuscript. It gives us great pleasure to know that the reviewer read our manuscript with great interest and found our methodology and statistical analysis to be accurate and exhaustive, including the Discussion and Conclusions of the study to be clearly and comprehensively presented. We have addressed the reviewer’s concerns point-by-point below. The reviewer’s comments are in bold and our replies are in italics. We hope the reviewer finds our replies and revisions to be satisfactory.

Herein I report only minor concerns:

Point 1: There are some typos and an English revision is required; please check the text again

Response 1: We thank the reviewer for their suggestion regarding typos and English language revision. We have now corrected the typos and revised the manuscript to improve the English language.

Point 2: In line 23, the acronym ATA is not made explicit previously in the text

Response 2: We thank the reviewer for their suggestion. We have now explicitly mentioned the full form of the acronym ATA at its first occurrence (Page 2, Line 78).

Point 3: In line 70, the acronym ETE is not made explicit previously in the text

Response 3: As suggested by the reviewer, we have now explicitly mentioned the full form of the acronym ETE at its first occurrence (Page 2, Line 80).

Point 4: To give uniformity to the text and make it easier to read, the Authors must report all the data in Arabic numbers, instead of spelling them.

Response 4: Respecting the reviewer’s suggestion, we have now reported all the data in Arabic numbers in the revised manuscript to give uniformity to the text and make it easier to read (Page 2, Line 63 and Page 5, Line 163).

Point 5: In my opinion, the manuscript could be enriched by these articles:

- DOI: 10.3390/jcm10184076, to be inserted in the introduction, where the references 1 and 2 are present

- DOI: 10.3390/cancers13215567, to be inserted in the introduction, where the references 17 is present

- DOI: 10.1371/journal.pone.0121514, to be inserted in the introduction, where the references 10 is present.

Response 5: We thank the reviewer for their valuable suggestion to incorporate additional references to enrich the manuscript. As suggested by the reviewer, we have now added these references to the revised manuscript.

Round 2

Reviewer 1 Report

I appreciate the authors' response to my comments. 

Yet, even the revised manuscript does not focus on the middle-east population uniqueness.

Your abstract and most of your introduction need to be revised in order to focus the reader on why you chose to focus the results on the middle-east population and how it is different than other Western countries. I would also suggest modifying the discussion to reflect the added value of this paper in comparison to much larger series in Western and far-East - japan, korea countries. 
